# Monitoring Methods of Human Body Joints: State-of-the-Art and Research Challenges

**DOI:** 10.3390/s19112629

**Published:** 2019-06-10

**Authors:** Abu Ilius Faisal, Sumit Majumder, Tapas Mondal, David Cowan, Sasan Naseh, M. Jamal Deen

**Affiliations:** 1Department of Electrical and Computer Engineering, McMaster University, Hamilton, ON L8S 4L8, Canada; faisaa4@mcmaster.ca (A.I.F.); majums3@mcmaster.ca (S.M.); sasan_naseh@yahoo.com (S.N.); 2Department of Pediatrics, McMaster University, Hamilton, ON L8S 4L8, Canada; mondalt@mcmaster.ca; 3Department of Medicine, St. Joseph’s Healthcare Hamilton, Hamilton, ON L8N 4A6, Canada; cowand@mcmaster.ca

**Keywords:** wearable sensors, joint monitoring system, joint angles, range of motion (ROM), skeletal tracking, goniometer, optical sensors, textile-based sensors, inertial measurement unit (IMU), sensor fusion

## Abstract

The world’s population is aging: the expansion of the older adult population with multiple physical and health issues is now a huge socio-economic concern worldwide. Among these issues, the loss of mobility among older adults due to musculoskeletal disorders is especially serious as it has severe social, mental and physical consequences. Human body joint monitoring and early diagnosis of these disorders will be a strong and effective solution to this problem. A smart joint monitoring system can identify and record important musculoskeletal-related parameters. Such devices can be utilized for continuous monitoring of joint movements during the normal daily activities of older adults and the healing process of joints (hips, knees or ankles) during the post-surgery period. A viable monitoring system can be developed by combining miniaturized, durable, low-cost and compact sensors with the advanced communication technologies and data processing techniques. In this study, we have presented and compared different joint monitoring methods and sensing technologies recently reported. A discussion on sensors’ data processing, interpretation, and analysis techniques is also presented. Finally, current research focus, as well as future prospects and development challenges in joint monitoring systems are discussed.

## 1. Introduction

The human body is a well-developed mechanical structure. The skeleton of the human body is made up of 206 differently shaped bones, which serves as a framework of the body, and the joints are the locations where bones meet each other. Joints hold the bones together and give the skeleton stability and mobility [1]. As we grow older, joints begin to deteriorate due to wear and tear as well as disease states. Three types of joints are present in the human body: fibrous (immovable), cartilaginous (semi-movable) and synovial (freely movable) joints [2]. Synovial joints (Figure 1a) are the key joints of our body because they provide mobility by allowing load-bearing, low-friction, wear-resistant smooth movement between articulating bone surfaces [3]. We have six groups of synovial joints in our body. These are categorized by the apposing bone surface at joints and the types of movement they permit: pivot, hinge, saddle, plane, condyloid and ball-and-socket joints [3]. These are presented in Table 1 and illustrated in Figure 1b.

All synovial joints of the human body are bound by a complicated system of ligaments, muscles, tendons and cartilage [4]. There are protective membranes and synovial fluid which lubricate those joints to facilitate smooth movements and load bearing [5]. Throughout our life and for every functional activity, these joints are critical and several bear our weight and are key to our movements. With ageing, synovial fluid production is reduced, cartilage wears, and the articulating bones come into direct contact, causing irregular articular surface and loss in bone density which are commonly known as musculoskeletal damage. Pain, stiffness, deformation, inflammation and swelling in the joints are the signs of this musculoskeletal damage. Worldwide, musculoskeletal disorders have become a serious threat to healthy aging. Musculoskeletal disorders are one of the major causes of work loss and early retirement, lost retirement wealth [6] and reduced productivity [7]. This disorder is the second most common cause of disability [8]. Persons with these disorders have significant morbidity and higher mortality rates than their age- and gender-matched peers [9]. When considering death and disability together, musculoskeletal disorders rank fourth (6.7%) in total global death and disability burden [10]. According to the Global Burden of Disease Study 2010 (GBD 2010), the impact on global disability burden due to musculoskeletal conditions is enormous, and it is the top-ranked common cause of disability among older adults. The proportion of years lived with disability (YLDs) due to musculoskeletal disorders is significantly higher while considering older age groups (50 years and more) in both developed and developing countries (Figure 2), and are predicted to increase dramatically in the coming decades [8].

Among musculoskeletal disorders, arthritis is one of the most common, and it is a major contributor to the world disability burden. From 1990 to 2013, it increased by 75% across the world’s population [10,11]. There are more than 100 different types of arthritis [11]. It can affect virtually any joint [12], and people of all ages, genders and races can have arthritis-related problems. Eventually, this leads to physical impairment and loss of mobility by making the affected joints very hard to function. According to the Arthritis Foundation [11], about 54.4 million adults in the U.S. (22.7 percent of all adults) had formally diagnosed arthritis and among them, 23.7 million (43.5 percent of those with arthritis) had arthritis-attributable activity limitation such as the inability to do daily activities (e.g., walking or climbing stairs). Losing these kinds of mobility has severe physical, mental and social consequences among older adults. Poor mobility gradually leads to a lack of independence, depression, reduced productivity, weakened ability in handling daily activities, and worsening health related quality-of-life [13]. The cost for direct treatment and healthcare services are a major social and financial burden on society and caregivers. Also, costs due to lost productivity (the indirect economic loss to society) outweigh the direct costs by a factor of five [14]. Responding to this increasing socio-economic burden demands a multilevel, integrated response, including primary prevention, early detection and effective intervention for persons at risk with common musculo-skeletal health issues.

Common factors that result in loss of mobility include aging, poor or infrequent physical activity and/or poor diet that leads to obesity and several chronic diseases. A smart wearable monitoring and assistive device can help a person by regularly monitoring the mobility status and assisting to maintain regular physical activities/exercises [15]. Such wearable devices that provide continuous monitoring of joint activity and health can record and extract important parameters for early diagnosis, leading to early treatment of mobility-related problems. Analysis of human joint posture and movement is fundamental for an extensive range of mobility-related activities such as rehabilitation, sports medicine, human activity assessment and virtual guided training [16]. To perform such an analysis, a complete set of data related to mobility and musculoskeletal health status is required. Then, mobility-related parameters from the processed data can be interpreted by the medical experts to provide useful information about the overall mobility status of the individual. Also, by comparing the measured parameters of an individual person with the reference values, accurate diagnosis of mobility-related problems becomes feasible. There are several sensing technologies capable of detecting joint parameters and movements. Most existing joint monitoring systems are large and complex, with a sophisticated and fragile structure [17,18,19]. Some systems also require a skilled operator and laboratory setup to gather the data and extract joint-related information. Fortunately, with advances in sensor technology, the monitoring process has become easier, more convenient and less costly to implement. Also, wearable sensors are now very reliable, and so are extensively used for healthcare, entertainment, security and consumer applications [20,21,22].

Wearable sensor-based health monitoring devices can monitor and record real-time information about one's physiological condition and motion activities [23]. These monitoring systems are comprised of a variety of miniaturized sensors that can be integrated into comfortable garments or directly attached to the human body [20]. There are several wearable sensors which are capable of measuring different physiological signals such as body temperature, heart rate (HR), electrocardiogram (ECG), electromyogram (EMG), respiration rate (RR), blood pressure (BP), arterial oxygen saturation (SpO2) and electrodermal activity (EDA) [20]. In addition, micro-electro–mechanical system (MEMS)-based motion sensors (e.g., accelerometers, gyroscopes and magnetic field sensors) are widely used for activity monitoring [19,20,21,22,24]. By combining those sensors with actuators, wireless communication modules and signal processing units, different viable real-time wearable health monitoring systems could be developed for various applications such as activity monitoring, fall detection, gait pattern and posture analysis, in sleep assessment, or early detection and diagnosis of several cardiovascular, neurological and pulmonary diseases. These systems can also raise an alarm in the case of any potential health issues and transmit the measured data to the healthcare personnel or services via a secure wireless media such as the internet or a cellular network, thereby functioning as the gateway to remote healthcare facilities. Thus, advanced sensors have opened the door of opportunity to develop a miniaturized wearable joint monitoring device which is accurate, durable and able to connect wirelessly with smart devices for easy, fast and seamless operation [24]. However, the selection of the sensor is critical, and depends upon several associated factors related to performance, cost, calibration and service of the entire system. To implement a comprehensive joint monitoring system, we need to integrate sensors, a data transmission device and a feedback system.

Various types of joint monitoring devices based on different sensing techniques and algorithms are suggested in literature [17,25,26,27,28,29,30,31,32,33,34,35,36,37,38,39,40,41,42,43,44,45,46,47,48,49,50,51,52,53,54,55,56,57,58,59,60,61,62,63,64,65,66,67,68,69,70,71,72,73,74,75,76,77,78,79,80,81,82]. The main focus of many researchers is to make the system simple, easy-to-use, cost-effective, non-invasive, unobtrusive and wearable with wireless communications [77]. With these features, the system can be used in real time for monitoring and analyzing the continuously collected data based on detailed input, requirements and specifications of the person being monitored [15,23,83,84,85]. In this paper, we present a detailed survey of different proposed and developed technologies and methods of joint monitoring. Three key parameters—joint angle, motion and skeletal tracking—for joint monitoring and their measuring techniques are discussed in the Section 2. Various types of sensors and technologies commonly used to develop joint monitoring systems are explained and compared in Section 3, highlighting their working principles, measurement parameters, data gathering and processing, and validation techniques. A short list of different types of published sensors and technologies and monitored joint parameters are shown in Figure 3, while Table 2 presents a comparison among them, mentioning the measuring techniques, method of analysis, their advantages and limitations. In addition, we also review several proposed sensor fusion methods for developing joint monitoring systems. Data processing, interpretation and proper analysis are also important to retrieve useful joint health information from the system data. Therefore, a brief discussion on data processing and analyzing techniques for further feedback and prediction application is presented in Section 4. Finally, in Section 5, we conclude the manuscript by summarizing the importance of joint monitoring systems with some key challenges and future research opportunities.

## 2. Key Parameters for Joint Monitoring

The physiologic joint movement occurs through the isotonic contraction of the muscles and the contractile force is closely related to the change of muscle length at the joint location [86]. During the contraction, the magnitude of muscle length change varies with different joint angles, motions and postures. By measuring the range of motion of a joint, it is possible to determine the maximum force generated by the muscle which represents the health condition of a joint. Similarly, the joint angles and postures at different activity levels also depict muscle strength and endurance [87]. Therefore, the principal focus of developing a joint monitoring system is to track and record the joint activities in the form of meaningful data such as angle, range of motion (ROM), motion, and orientation, which could potentially be used to estimate the joint health status and provide feedback to the person being monitored. For measuring and assessing the mechanics of human joints in a variety of activities, different tracking techniques are being used. These techniques can be categorized based on three key extracted parameters: joint angle, joint motion and skeletal tracking.

### 2.1. Joint Angle

Each movable body joint has an optimal joint angle range for a specific activity or motion. To reach that optimal angle, the muscle should have the correct length to bear the maximum strength [88]. There are several published reference values of the active range of joint angles or range of motions [89,90,91] for a healthy adult. However, the ROMs vary depending on sex, age, physical structure, daily activities, etc. [92]. In Table 3, the normative values of normal joint range of motion (ROM) in 674 normal subjects by gender (54% females and 46% males) and four different age groups (2–8, 9–19, 20–44 and 45–69 years) measured in degrees [93] are presented. Figure 4 shows a pictorial view of the five joints’ movements. These reference values were calculated along with 95% confidence intervals for normal range of motion for 11 different movements measured on five joints.

From Table 3, for all joints, a downtrend of ROM values is visible with aging, for both male and female subjects. The greatest change was seen in knee flexion, with a 15° difference in mean ROM between the age groups of 2–8 years and 45–69 years. Although ROMs are generally affected by aging, there can be other reasons such as injuries or other health-related problems which can cause a reduction in the ROM of a joint. Therefore, it is very important and useful for therapists and physicians to study joint angles for the early detection of joint issues or determine the progress in joint rehabilitation. For joint angle measurements, several sensing systems were proposed and developed [26,42,43,44,48,49,50,51,52,53,63,66,67,68,69,70,71,72,94,95]. Most joint angle measurement systems were based on mechanical or electromechanical goniometers that used resistive potentiometers or strain gauges [17,29,96,97]. However, the major disadvantages of these systems were inflexibility and lower accuracy. To overcome these drawbacks, optical-based goniometers were developed using the optical properties of the sensor to calculate joint angle [17,29]. For example, some systems are based on optical fiber sensors which measure the attenuation of the transmitted optical signal power which is correlated with the bending angle of the fiber [25,26,27]. Other systems used textile-based conductive wire sensors or flex sensors in which the joint angle is proportional to the change in the sensor’s resistance [37,38,39,40,41]. However, compared to other sensor technologies, inertial measurement unit (IMU) sensors are now being increasingly used to measure the joint angle. This is because IMUs have small size, are low cost and can measure 3D angles with high precision and high accuracy [62]. An IMU is a compact device combining three different sensors: accelerometer, gyroscope and magnetometer. Inertial measurement unit sensors are now widely used in wearable devices for monitoring and measuring joint angles [62,63,65,66,67,68].

### 2.2. Joint Motion

The normal motion of a joint describes its movement from the center of the joint location [98]. Joint motion includes flexion (bending), extension (straightening), adduction (movement towards the center of the body), abduction (movement away from the center of the body) and rotations (inward and outward movements) [99]. The range of motion (ROM) value denotes the full movement potential of a joint (Table 3). Because of some health issues or injury, a joint may experience reduced ROM [99]. Therefore, ROM is a useful clinical indicator in evaluating the health status of a joint. Also, for joint rehabilitation, continuous motion monitoring is very important. The measurement of a joint motion includes both its angle and orientation. Therefore, joint monitoring systems based on goniometer [17,29], optical fiber sensors [25,26,27] or flex sensors [37,38,39,40,41] cannot be used to monitor joint motion since they can only measure single-axis movement which is interpreted as an angle. To estimate both the angle and orientation of joints, the most common technique is imaging or video-based tracking where the visual data of several human joint actions are captured to estimate joint motions using anthropometric constraints (size, shape and composition of the human body) and known joint locations in reference images/videos by applying image processing techniques [32,33]. Although this system is a reliable and well-established monitoring technique, it needs a pre-equipped environment and setup which does not allow for continuous and long-term monitoring, especially during normal daily activities. On the other hand, a 3D IMU sensor is able to measure the 3D motion of joints [63,64,65,66,67,68]. The IMU’s miniature size, low-cost and ease-of-use makes it suitable for a wearable, low-cost, continuous joint monitoring system.

### 2.3. Skeletal Tracking

Skeletal tracking is a technique used to build a skeletal model of a human body by detecting the positions of various joints on a human form [100]. Skeletal tracking helps to detect real-time human pose by realizing and understanding the human posture for different functional activities [34]. The key applications of skeletal tracking are to detect physical disability and analyze rehabilitation process by assessing the body posture accurately [101,102,103,104,105,106,107]. Most skeletal tracking systems are based on image processing techniques using a single camera [34] or a network of multiple cameras to capture images in different activities and covering a large area [32]. This technique includes the quantitative measurement of the positions and movements, complicated image processing algorithms to analyze the depth of image, and machine learning to track human joints and create the human model [100]. Instead of using cameras or imaging sensors and complicated image processing techniques, some researchers are replacing such imaging systems with multiple calibrated IMU sensors placed on different joint locations on a human body for skeletal tracking [78,108,109], and this also make the IMU-system more convenient and easier to use. Currently, skeletal tracking is widely used in several smart activity monitoring systems, human-computer interaction, surveillance, virtual gaming, pattern recognition, human behavior detection, athlete performance analysis and rehabilitation systems [18,32,33,34,35,36,49,77,83].

## 3. Sensors and Technologies

Sensors are the fundamental elements of a joint monitoring system to measure the physiological parameters—angle, range of motion and posture—of joints. Various research groups [16,17,25,26,27,28,29,30,31,32,33,34,35,36,37,38,39,40,41,42,43,44,45,46,47,48,49,50,51,52,53,54,55,56,57,58,59,60,61,62,63,64,65,66,67,68,69,70,71,72,73,74,75,76,77,78,79,80,81,82] have used different sensor types to build joint monitoring systems. Some systems were developed using a single sensor; others were assembled with a combination of multiple sensor technologies using data fusion methods. The selection of sensors is crucial when developing an accurate and reliable monitoring system. The key features to be considered while developing such a system are high efficiency and accuracy, good reliability, high sensitivity, small size, light-weight, lower energy consumption, and low processing resources [24]. In this section, we will discuss various sensor technologies used to develop joint monitoring systems.

### 3.1. Optical Sensors

Most of the common implementations of optical sensor-based joint monitoring system used either intensity modulation or optical navigation methods [17,26,29,110]. Optical fiber sensors (OFS) were used for intensity modulation [25,27,111]. The basic working principle of OFS-based systems is to detect the attenuation of the transmitted optical signal power due to bending of the optical fiber [27]. On the other hand, the systems using optical navigation sensors detect planar motion with joint movement. The optical navigation-based monitoring device is also known as optical-based goniometer [17,29,30].

Optical fiber sensors are now often used for designing various health monitoring systems [25,111,112,113,114], and several categories of OFS were developed for both health-related academic research and commercial products. Depending upon the measuring techniques of the physical parameters and their conversion methods from optical data, OFS can be categorized into four different types: single-point, long-gauge, quasi-distributed and distributed sensors [25]. Optical fiber sensors are made of flexible plastic optical fibers through which optical signals are transmitted. The basic components of an OFS-based system are a light source, flexible optical fiber and a photodetector. The light source generates the optical signal that travels through the flexible optical fiber and is received by the photodetector at the end of the fiber. By measuring the attenuation of the optical signal, it is possible to determine the bending angle of the fiber [26]. Due to this simple sensing principle and structure, optical fiber sensors can be easily integrated into a monitoring system for measuring human joint angles [111]. Basic configurations and working principles of an optical fiber and OFS-based joint monitoring system are shown in Figure 5a,c.

The main benefits of OFS are high resolution, flexibility, light-weight and immunity to electromagnetic interference. Different techniques were developed to improve the sensitivity and accuracy of OFS-based joint angle measurements [26]. For example, roughening or polishing the surface in one side [28] of an OFS is a common method to improve the sensitivity by enhancing the optical signal attenuation with bending [115]. In the work by L. Bilro et al. [26], a wearable knee motion monitoring system using a flexible plastic OFS placed in a commercial knee brace was developed. The change of transmittance (the ratio of the light energy incident on an object to that transmitted through it) was measured while bending the plastic optical fiber. One side of the fiber was polished to improve the performance as macro-bending creates more attenuation [110,116]. This device was made wearable and wireless by integrating a wireless communication board based on Bluetooth technology with the main controller board. A comparative analysis with a reference video-based monitoring system was made and the average deviation of angle measurement between the two systems was 2.1°.

Another optical fiber sensor-based human joint monitoring system is presented in [27]. In this system, the authors used a fiber-optic curvature sensor with a different sensitive zone configuration and the diameter of the configured optical fiber was 1.5 mm. A “teeth-like” configuration was created by drilling precisely on one side of the fiber [117] to make the sensitive zone (see Figure 5b). Due to this sensitive zone, when the optical fiber bends while keeping the teeth on the convex side, the light intensity on the outer side decreases. Conversely, the light intensity increases when the teeth are on the concave side. LabVIEW software and ZigBee-based wireless communication were used to complete the system. The optical intensity was exponentially dependent on the curvature angle and the angle range of the system was −120° to 120° with a 1 Hz sampling rate. However, a linear characteristic was found between −45° and 25°, with an average sensitivity of 20 mV/° (voltage change per angle) and a resolution of 1° [27]. The sensor had a high operating temperature limit (up to 70 °C) without any deformation or characteristics change. Therefore, within certain ROM limitations, the proposed sensor was suitable for developing a low-cost and simple wearable joint monitoring system with wireless communication capabilities.

The goniometer is one of the most commonly used instruments in human joint related research and clinical monitoring [17,29,96,97]. Goniometers can be used to determine the range of angular motion of different human body joints such as the knee, elbow, or waist [29]. Most goniometer-based joint monitoring systems use mechanical or electromechanical goniometers, which are based on resistive potentiometers or strain gauges [17]. The main disadvantages of these kinds of goniometers are large size, imprecision, flimsiness and fixed center configuration which does not provide flexibility with the natural joint movement [97]. To overcome these difficulties, some researchers have proposed optical-based goniometer systems for joint monitoring, as they are flexible, unobtrusive and of higher precision [17,29].

An optical fiber-based goniometer presented in [17] was made of a single-mode optical fiber. Intensity modulation of a propagating laser beam was used to detect the changes in polarization due to the rotation of adjacent portions of fiber. Controlled birefringence was induced by a fiber loop with a fixed radius. The components of the reported goniometer and its working principle are shown in Figure 6. A trans-impedance amplifier with a high gain-bandwidth product gave a high-precision output signal with high sensitivity. A sample-and-hold circuit, an acquisition board and a computer-based software program were designed to gather and process the data. The goniometer was combined with a fabric to build a flexible, compact and accurate wearable joint monitoring system with several applications such as testing an athlete’s performance and training status.

In [29] another wearable system using optical-based goniometer for joint monitoring was proposed. These authors used a technique similar to the optical mouse which has a small camera to identify two-dimensional planar motion by detecting the displacement. They chose the elbow joint for their experiments and their system consisted of two units: hardware and firmware. The hardware unit had three components: the sensor, microcontroller and the joint module (a flexible strip). The firmware unit was for the communication and data gathering. The proposed sensing system setup for elbow joint measurement is shown in Figure 7. The flexible strip was placed around the joint with one end fixed, and the sensing unit was placed on the other end. The sensing unit was able to move freely along with the strip and measured the uniaxial displacement during bending of the elbow joint. The angle (*a*) was calculated from this linear displacement (Δ*x*) that is directly correlated with the bending (Equation (1)).
(1)$$\alpha =\;\frac{\Delta x}{R}\cdot \frac{360{}^{\circ}}{2\pi }\;$$

The joint radius (*R*) was assumed to be constant for this calculation. The proposed sensing module is light-weight and easy to assemble as a joint monitoring system. However, the system can only monitor one-dimensional movement which can affect the angle measurement accuracy for human body joints.

### 3.2. Imaging and Video-Based Tracking System

Imaging and video-based human skeletal tracking is a well-accepted method for human joint monitoring because of its broad applicability and reliability [31,32,34,35,36,101,102,103,104,105,106,107]. One or more cameras are the core components of this system. The process flow of this method is to capture the visual data of several human actions by using a single [34] or multiple camera network [32], and then track the joints using anthropometric constraints (size, shape and composition of the human body) and known joint locations in reference images/videos with the same action. These applications comprise various fields of research such as biomechanics, image processing, machine learning and pattern recognition [33]. The main challenge of this system is to construct a three-dimensional human model using a single static camera. A new image processing method was proposed in [118] where they used 2D images to create a 3D model. They introduced a new image descriptor based on discrete cosine transform (DCT), which was used in the pose-matching procedure for finding appropriate action in the reference database using an interpolation and tracking process. The descriptor matrix was divided into three frequency regions for different levels of tracking: (1) low-frequency region containing the general shape and intensity information of the joint; (2) middle-frequency region with general edge information; and (3) high-frequency region consisting details of the tracked joints. Both discriminative and tracking algorithms were used in this method to increase joint tracking accuracy. A block diagram of an imaging-based human skeletal tracking is presented in Figure 8.

A single camera-based system can only detect the joint location within its field of view, thus limiting the range of observations. To solve this problem and track human joint motion in a large area with multiple fields of view, a distributed camera network system was suggested in [32]. They set up multiple cameras to make the network and used an information-weighted consensus filter (ICF) as a distributed estimation algorithm to track human motion in a camera network inside the sensing range. The distributed camera network setup is shown below in Figure 9a. They used the Microsoft Kinect image sensor instead of a usual camera to build the camera network [32] because Kinect can measure the joint locations without any markers [34,67]. Kinect is a high-end imaging device with an RGB camera, a multi-array microphone and a built-in laser projector combined with a monochrome CMOS sensor that make it capable of capturing color images and depth images (Figure 9b). In addition, it has a skeletal tracking tool which is able to recognize 20 different joints’ locations of a human body. Therefore, instead of a usual camera, Kinect has been chosen by many researchers who are working on vision-based human joint monitoring and analysis problems [31,32,34,35,36].

Although imaging and video-based joint tracking system is a popular and reliable monitoring technique [31,32,34,35,36,101,102,103,104,105,106,107], it requires complex, expensive infrastructure and sophisticated analyses of data-intensive video streams. Also, this system is only effective with a pre-equipped environment and setup, restricting users’ usual movements, which makes it unsuitable for continuous and long-term joint monitoring in daily activities.

### 3.3. Textile-Based Sensors

Textile-based sensors (e.g., flexible conductive wire sensors, flex sensors, strain sensors, etc.) are very suitable for developing a wearable joint monitoring system. The working principles of all these sensors are similar. In all cases, changes of resistance are measured, and these changes are directly related to the corresponding joint angles [38]. To develop a long-term and regular wearable monitoring device, textile-based sensors can be a good choice because of their flexibility and simple sensing principle. Furthermore, they can be easily integrated into stretchable skin-tight fabrics around the joints [37,38].

A flexible conductive wire sensor-based method was proposed in [37] where the authors incorporated flexible conductive wires in flexible and comfortable fabrics for joint monitoring. They implemented a single-axis arrangement with a single conductive wire designed for the knee joint (see Figure 10). The parameter measured was the resistance changes of the conductive wire with the movements of the joints. Figure 10 shows the implementation of the system where one end of the flexible conductive wire was fixed to the fabric above the knee at point 1 and the other end was connected with an elastic cord at point 3. The elastic cord was attached to the fabric below the knee at point 4 and helps the conductive wire slide freely with bending and stretching. There was a wire contact point between point 2 and 3 along the fiber and it was permanently stitched into the fabric at point 2 to keep it fixed. The fiber slides with the movement of the knee, causing the change of conductive thread length between 2 and 3 as well as the resistance in that portion [37]. The resistance value at a specific point is directly proportional to the knee angle. They also proposed a multiple conductive wires-based system to monitor multi-axis joint (e.g., hip and shoulder) angles. Their future focus in this research is to improve the accuracy of the system by applying a more precise technique of incorporating conductive wires into flexible, skin-tight fabrics and adding a wireless module for data transmission.

Flex sensor is another type of textile-based sensor which is usually made of a conductive material with flexible and stretchable properties [38]. The shape of the sensor will change with the applied force, causing the resistance change between two measuring points. Therefore, the flex sensor is convenient for wearable joint monitoring systems by integrating it with comfortable garments [38,39,40,41]. The flex sensor is usually stitched to the flexible and skin-tight garment across the joint to be monitored. Whenever the joint bends or stretches, the pressure changes on the sensor which causes the variation of its electrical properties (resistance). This resistance variation due to the joint movement can be measured using an electronic system to quantify the joint angle [40]. Textile-based highly stretchable strain sensors are also used by some researchers for human joint monitoring [43,44,45,46,47,48]. Conductive yarns were employed as the conductive part of the strain sensor. Different textile materials with preferable elasticity and conformability were used to fabricate the system comfortable for human joints [44,45]. Another type of textile-based sensor was developed in [48] by using flexible and stretchable CCF (chopped carbon fiber)/PDMS (polydimethylsiloxane) conductive yarns. The CCF/PDMS composite sensors were integrated into the textile structures and used the piezoresistive (resistance-strain) behavior of the sensors for detecting human joint motion.

### 3.4. Inertial Measurement Unit (IMU) Sensors

An IMU is a combination of three sensors (accelerometer, gyroscope and magnetometer) and is used to measure the three-dimensional acceleration, angular velocity and the magnetic field vector in their own coordinate systems. As a unit, the three sensors are calibrated in such a way that each of their individual coordinate system acts as an orthogonal base which typically remains well aligned with the outer casing of the unit [62]. Moreover, there are some commercially available IMU sensors with built-in algorithms to fix the sensors’ orientation with respect to a global fixed coordinate system (e.g., [119]) which can be represented by a rotation matrix, a quaternion, or Euler angles. For developing a wearable measurement system for human joint motion, IMUs are the most promising and compact devices for both clinical assessment and research studies, because of their small size and capability to measure joint motion with precision and accuracy [62,63,64,65,66,67,76]. To detect position and orientation, three-dimensional angular velocities and linear accelerations are measured using the IMU sensors. Most of the IMU-based joint monitoring systems use two calibrated IMUs placed below and above a joint [62,63,64,65,66,67]. Relative data from those two IMUs are compared for tracking the joint angle and motion.

For accurately computing a joint angle, we need to compensate for joint alignment using the two IMUs method described in [64]. These authors proposed a manual method using a set of predefined postures of different leg movements to align two IMUs attached on the thigh and shank. A fusion algorithm was then applied to measure the 3D knee joint angle. The measurement was then validated against the Liberty magnetic motion capture and tracking device (Polhemus, Vermont, USA). Later they added a functional calibration procedure which only relied on the IMUs data and made an error assessment (Table 4) by comparing the results obtained from the combined method to the reference system [68]. This method can be applied to monitor complex joints e.g., knee, ankle or elbow. A similar system was developed in [63] where Bluetooth technology was added for wireless communication. The system was evaluated by comparing it with an infrared motion capture system having an average deviation range of 0.08° to 3.06° from each other. Figure 11 illustrates two IMU sensors-based configuration for measuring the knee angle (α).

In [69] two IMUs-based joint angle measurement methods were presented. There, they transformed the measured data from both sensors into a joint coordinate system by aligning the IMUs’ local coordinate axes with the joint axis. Their first approach included the magnetometer readings (magnetic field vectors) to get a precise alignment. In the second technique, they relied only on the accelerometer and gyroscope data. Indoor measurements can suffer from magnetic disturbances due to other magnetic devices that may be present. Therefore, the authors excluded the magnetometer data for indoor monitoring and achieved equally accurate results. Thus, they were able to increase the accuracy of joint angle measurements by two IMUs having a precise calibration and alignment technique.

An IMU-based auto-calibration method was proposed in [71]. First, the limitations of an existing position calibration method were identified by performing a theoretical analysis (evaluation of observability by computing the Fisher information matrix). Based on that analysis, a new method to continuously determine the IMUs’ relative position with the joints was introduced. Then, based on the simulated and captured data, an experimental evaluation was performed to present the enhancement of the calibration method. In [67], another IMU calibration and alignment protocol based on simulation and experimental analysis was proposed. These authors simulated a computer-based lower body anatomical frame with four IMU sensors to estimate the angles of hip and knee. In the simulation, the sensors were placed on the pelvis, right thigh, right shank and right foot, and aligned with the associated limbs’ coordinate system. They also made a joint model using two semi-spheres interconnected by a universal goniometer and placed IMU sensors on each sphere. They used this model to evaluate the accuracy and repeatability of the system while measuring angular movements. Finally, they performed a real gait test involving five healthy volunteers and validated the method by comparing the results from experiment and simulations.

The drift effect is another concern when using IMU sensors to calculate joint angles and estimate orientations. A random bias drift which builds up over time, affects the sensors tracking accuracy [70]. A new kinematic model was proposed in [70] to minimize this drift error. There [70], they considered natural physical constraints (age, gender and bone structure) while measuring the range of motion for each joint with their system. They modeled the sensor’s random drift and used zero-velocity updates. To avoid the complex linearization process, they implemented an improved version of extended Kalman filter (EKF) which is called unscented Kalman filter (UKF). Instead of estimating nonlinearity, it approximates the distribution of the measured data. They validated the algorithm by comparing their inertial tracker’s result with a reference optical tracking system and a high-precision industrial robot arm.

Instead of conventional two IMUs-based joint monitoring system, a single IMU-based system was developed in [72] to monitor hip and knee joint angles. The single IMU was placed on the shank and the collected data was utilized to estimate 3D lower-limb (pelvis, thigh and shank) joint kinematic quantities during five different lower limb motions with the help of the least-squares identification algorithm. It achieved an average accuracy of 3.2° and a correlation coefficient above 0.85 by comparing with reference data from a stereophotogrammetric system. One limitation of this method was the degradation of joint angle estimation due to the IMU’s motion artifact. Moreover, no pelvic motion was assumed in this approach. Thus, the quality of the result might be reduced with large pelvis movements.

By using an IMU-based system, we can measure not only joint angles, but also other important gait parameters e.g., cadence, step length, step variability, lateral and vertical excursion of the center of mass, etc. [66] for gait analysis. IMU sensors are very convenient to develop such a system because of their miniature size, low-cost and flexibility to use without space restrictions compared to traditional gait analysis methods such as semi-subjective techniques, imaging or floor sensors [120]. Moreover, most modern commercial IMUs have an integrated wireless module which makes it more appropriate to develop a wearable system for continuous joint monitoring and gait analysis [65,66,73,77,78,79,121,122,123]. A fundamental process flow of measuring joint angles using IMU devices (adapted from [65]) is described below in Figure 12.

A list of IMU sensor-based joint monitoring techniques, analysis and validation methods are summarized below in Table 4.

In [60], the authors used only gyroscope sensors to measure human joint angles. Gyroscopes in modern IMUs can measure three-axes angular rate with movements. For measuring the joint angle, two gyroscopes were placed above and below the joint location and calibrated before joint motion. Movement angles of each gyroscope were calculated by integrating the angular rate. Then, the joint angles were extracted by comparing the changes in angle between two gyroscopes using trigonometric functions. Two optimization filters: Median filter [60] (a non-linear digital filtering technique to remove noise from the sensor signal) and Kalman filter [60] (an algorithm that uses a series of measurements gathered over time, having statistical noise and other inaccuracies, and produces estimates of unknown variables that tend to be more accurate than those based on a single measurement alone, by estimating a combined probability distribution over the variables for each timeframe) were used to mitigate the noise and drift in the sensors to yield optimized output [60,124].

Some other research groups have proposed using only a magnetometer-based sensing system using magnetic fields to determine movement. A wearable field generator design was introduced in [125] to build a fingertip glove equipped with magnetic tracking sensors. Another magnetic sensor-based hand glove in [126] was designed with 20 Hall-effect sensors embedded. This glove was used to assess hand orthopedic disorders by detecting the relative and absolute orientation of the fingers. In [61], a magnetometer-based nonobtrusive system for monitoring the wrist and hand movement was proposed. A magnetic (neodymium) ring worn on the index finger and two triaxial magnetometers mounted in a watch-like unit to measure the magnetic field produced by the magnetic ring and sent the data to a wireless device were used. The movement of the finger was then calculated from the reading of the magnetic field which was correlated to the finger motion. The accuracy of the proposed system was analyzed by comparing it with a traditional goniometer-based system. An average accuracy of 92%–98% with a 19%–28% standard deviation was obtained.

### 3.5. Sensor Fusion

Recently, some researchers have used sensor fusion methods (combination of multiple types of sensors) to develop more precise and reliable joint monitoring systems. Sensor fusion is the process of combining multiple sensor data in such a way that the combined output shows better performance than individual sensor results [127]. Thus, sensor fusion allows multiple viewpoints with improved resolution, greater spatial and temporal coverage, reduction in ambiguity, and greater precision in measurements. A simplified block diagram of sensor fusion (adapted from [127]) is shown in Figure 13.

In [75], a sensor fusion method combining both flex and gyroscopic sensors was proposed. Multiple flex sensors and a MEMS gyroscope were mounted on a supportive fabric worn by the subjects. To fuse the measurement by multiple sensors and estimate accurate joint angles, Kalman filtering was used. The authors built a behavioral model of joint movement over time and updated the system data using the model in Kalman filtering. The main purpose of using multiple sensors was to minimize sensor errors and reduce measurement noise.

Another method proposed in [16] was a fusion of textile electro-goniometer and accelerometer based on the Kalman filter. The purpose of this approach was to avoid pre-estimation of the accelerometer position and alignment. The focus was to measure knee angle during various motion activities and the system was standardized by comparing it with a commercial IMU-based system. Their technique used the data from the accelerometer to continuously adjust the goniometer reading and did periodic calibration. Thus, the fusion system showed more accurate angular measurement (RMSE: mean, μ = 1.96° and standard deviation, σ = 0.96°) compared to the individual derived estimation by the accelerometer (RMSE: μ = 6.55° and σ = 2.87°) or textile electro-goniometer (RMSE: μ = 5.15° and σ = 0.47°).

A study on data fusion from wearable IMUs and surface EMG sensors to monitor and assess human motor function, was presented in [128]. To estimate a motor function abnormality, a group of machine learning algorithms was used on the fusion data. To validate the algorithms’ effectiveness, two parameters: normal data variation rate (NDVR) and the determination coefficient (DC) were derived. A lower NDVR value represents better validity and a larger DC value represents a higher consistency and reliability of the system. Through experiments, these authors proved that the fusion result was superior to the sensors’ separate data: a reduced NDVR and a better DC from a regression analysis performed between the derived indicator and the routine clinical assessment score were obtained.

A fusion of data from optical sensors and inertial measurement units (IMU) to analyze human movement and design human kinetic energy harvesting systems was presented in [129]. High-speed cameras were used as optical sensors to determine the positions and angles of the joints. IMUs were used for acceleration, angular rate and magnetic field vector measurements. The fusion data was used to compute actual orientation and linear acceleration. The authors used this result to estimate the kinetic energy generated in different joints during several body motions. The purpose of this analysis was to design an energy harvesting system by converting human kinetic energy to electrical energy and then find the recommended joints (knees and ankles while walking) to place the energy harvesting module.

From the discussions above, the main purpose of fusing data from multiple sensors in a joint monitoring system is to overcome physical limitations of the sensing system and improve measurement accuracy as well as reliability. This is done by minimizing the error rate and enhancing the signal to noise ratio (SNR) while maintaining practical usability. In addition, sensor fusion offers several other advantages such as improved resolution, increased confidence in results, robustness against interference, and reduced ambiguity and uncertainty. Although it is in the early stages of development, current research results suggest that fusion systems are superior to other means of measuring and monitoring joint movement using one or a few sensors.

### 3.6. Other Sensors and Techniques

Some researchers have proposed acoustic emission (AE) sensor-based joint monitoring systems where they used piezoelectric-films or MEMS-based microphones to record the sound produced by a moving joint [55,56,57,58,59]. This acoustic emission from joints also known as vibroarthographic signals (VAG) are considered as clinically relevant biomarkers for joint health [58]. The researchers utilized the recorded signal to quantify the consistency of acoustic emissions from joints with respect to joint angle and position [57]. The emitted acoustic signal from an over-exercised joint during motion produces higher amplitude and shows a different pattern in the frequency domain compared to the healthy joints [55]. However, one of the major challenges is the background and interface noise that need to be removed to improve the signal to noise ratio (SNR) of the emitted signal [58].

A few research groups [49,50,51,52,53] have used different inbuilt smartphone sensors and cameras to measure joint angle and motion. Several applications (apps) were used to access and analyze the sensor data. These apps are mostly based on inbuilt smartphone sensors such as accelerometer [49,51], gyroscope [51,52], magnetometer [53] or camera [50]. In all cases, the results from the apps have shown adequate validity when compared against universal goniometer or inclinometer. In addition, a comparison study between two smartphone-based apps (inclinometer and camera) was published in [50]. In this study, the camera-based app provided higher precision and accurate measurement (a mean difference of <1° and 1/50 difference >3°) compared to the inclinometer-based app (a mean difference of <7° and 16/50 difference >10°). These researches suggest that the newly developed smartphone-based apps show potential as a useful tool for joint health monitoring [54].

Along with angle and motion, there are some other physiological parameters related to joints which can provide important information about joint health status. For example, changes in local skin temperature around the joint can be an indication of pathology. Local temperature change occurs due to the changes of blood flow in that region. Generally, the skin temperature of an inflamed joint is higher (1.1–2.8 °C) than a normal joint [130]. Including a temperature sensor with a joint monitoring device can be an effective add-on to the system. Measuring muscle pressure/force around a joint during several activities using flexible pressure sensors can also provide a correlation to the corresponding musculoskeletal health. Muscle pressure changes around a joint with every movement (flexion, extension, etc.) and the values are different for different activities. Muscle pressure plays a crucial role in determining the force balance, contact force and pressure distribution of the joint [131]. Any imbalance in muscle forces can result in joint pain and stresses. Flexible pressure sensors can be easily integrated with a joint monitoring system to measure the muscle forces around the joints. To estimate the muscle forces, electromyography (EMG) sensors can be used [132,133].

For joint monitoring, another important parameter is the sweat rate of the joint skin. It can provide necessary estimation about the thermal status and other physiological conditions of joints. Stress and perspiration in the joint are related to the local blood flow and the activities of sweat gland which can be interpreted by measuring the sweat rate. The galvanic skin response (GSR) sensor is one of the common types of sweat rate sensors. This sensor is used to measure skin conductance which varies with the change of skin moisture caused by sweating. A GSR sensor system has two components: the conductive electrodes to make contact with the skin and the electronic board to measure the skin conductance between two electrodes [134]. Another sweat rate sensor design concept proposed in [135] had two parts: the humidity chamber and the humidity sensors. Sweat-induced skin humidity is collected in the chamber and the humidity sensors are used to measure the humidity. Therefore, implementation of a monitoring device combined with these other relevant sensors would represent the first of its kind for joint monitoring, early diagnoses of joint related problems and quantitative information for joint health improvement.

## 4. Data Processing, Interpretation and Analysis

The purpose of a health monitoring system is well-served when the sensor data is securely stored and analyzed to retrieve necessary information for detection, prediction and diagnostic decision making [136,137]. For this reason, data processing techniques are a vital phase of health monitoring research. There are three major steps involved in building a complete joint monitoring system: (1) data acquisition from the sensors and data preprocessing; (2) feature extraction and selection; (3) modelling data by learning the extracted features (including expert knowledge and metadata) to perform the tasks such as detection, prediction and decision making [136]. A generic flow diagram of a joint monitoring system (adapted from [119]) is given in Figure 14.

Usually, the raw sensor data from a joint monitoring system is preprocessed to mitigate noise, motion artifacts and sensor errors. To remove motion artifacts and high-frequency noise, the data is passed through various types of filters [138,139]. Power spectral density (PSD) analysis and fast Fourier transforms (FFT) are the most common frequency domain analysis methods to detect the periodic fluctuations in sensor signals. Different filtering tools are then used to remove the noise. If it is a multi-sensor-based system, then the data gathered from different sensors need to be normalized and synchronized. Therefore, preprocessing also includes data formatting, data normalization and data synchronization, which are needed to handle the complex and large amounts of data from the sensors [140].

Feature extraction is the process of discovering the main features of a data set which represents the characteristics of the original data [141]. The extracted features are used to formulate the relationship between the key parameters discovered from the raw data and reference knowledge to prepare the model for prediction and decision making [142]. Features are extracted from analyzing signals in two different aspects: time domain and frequency domain [143]. In time domain extraction, basic waveform characters and statistical parameters related to the visible attributes in the data are the most common features such as maximum, minimum and mean value, median, standard deviation, slope, rise time and fall time, zero crossings count, peak value and duration of the sensor data [144,145]. In joint monitoring, these time domain features related to joint parameters are important because the conventional decision-making frameworks are based on the observable trends in the signal [146]. On the other hand, to get extra information about the periodic behavior of the sensor signal, more features such as power spectral density, mean frequency, peak frequency, spectral energy and wavelet coefficients of the signal are acquired from frequency domain analysis [140,147]. After extracting the features from the raw data, feature selection techniques are applied to select more discriminative features and reduce the dimensions of input data.

Feature selection methods can be classified into three types [141]: (1) wrappers, which use classifiers to score a given subset of features; (2) embedded methods, which inject the feature selection process into the learning of the classifier; and (3) filter methods, which analyze intrinsic properties of data, ignoring the classifier to rank the features and create feature subsets. Every feature selection technique can be also categorized as supervised or unsupervised learning [148]. In supervised learning, class labels are given beforehand, and the algorithms maximize some functions to select the relevant features which are highly correlated with the class. On the other hand, class labels are not specified in unsupervised learning. Therefore, it becomes more difficult when relevant features are needed to be found simultaneously.

Some commonly used feature selection methods are, the Fisher method [149], infinite latent feature selection (ILFS) [150], feature selection via eigenvector centrality (EC-FS) [151], dependence guided unsupervised feature selection (DGUFS) [152], feature selection with adaptive structure learning (FSASL) [153], unsupervised feature selection with ordinal locality (UFSOL) [154], least absolute shrinkage and selection operator (LASSO) [155] and feature selection via concave minimization (FSV) [156]. There are some other well-known feature selection methods such as principal component analysis (PCA), independent component analysis (ICA) and linear discriminant analysis (LDA) [157] which are used in health monitoring to reduce the dimensions of large physiological data sets. By using those feature selection tools, the subset of the most significant features is separated for modeling and learning methods. A category of different feature selection methods, their advantages and limitations are given below in Table 5 [158].

After completing preprocessing, feature extraction and selection steps, it is necessary to apply the proper analysis method to make sense of the data and retrieve meaningful information. Examples of such meaningful information include anomaly, outliers and alarms detection which are used for further prediction, diagnostic decision making and feedback applications [138]. There are several developed machine learning algorithms which are used to analyze the physiological data from different sensors based on their usability and efficiency. These methods can be classified into two types [159]:(1)Supervised learning in which a predictive model based on both input and output data is developed; and(2)Unsupervised learning which discovers an internal representation from input data only.

The algorithms related to classification (output is a choice between classes) and regression (output is a real number) are categorized as supervised learning. Popular examples of classification methods include support vector machines, discriminant analysis, naive bayes, nearest neighbor, rule-based methods and association rule mining. besides, linear regression generalized linear model, support vector regression (SVR), gaussian process regression (GPR), logistic regression models, ensemble methods, decision trees, artificial neural networks are examples of regression techniques. On the other side, all of the clustering algorithms (no output—find natural groups and patterns from input data only) are used as unsupervised learning. Some of the most common clustering algorithms are k-Means, k-Medoids, fuzzy c-Means, hidden Markov models, Gaussian mixture models, hierarchical clustering, etc. [136,160]. A comparison among the most common algorithms is summarized below in Table 6.

## 5. Conclusions and Future Challenges

The aim of this study was to present a state-of-the-art survey on different monitoring methods of human body joints which includes the key joint parameters, sensor technologies and the developed systems and their performance analyses. A comfortable, wearable and easy-to-use sensor-based joint monitoring device is a promising solution to the mobility and other musculoskeletal issues. In this paper, we provided some background on existing systems and the current trend of sensing technologies for joint monitoring. We then reviewed the key parameters and different sensors technologies published or commercially available for joint and activity monitoring which included optical fiber sensors, optical-based goniometer, imaging and video-based tracking system, textile-based sensors, gyroscope, magnetometer and inertial motion sensors. We discussed and analyzed their working principles, measuring techniques, validation process and performance compared to others. We also explored some sensor fusion methods which were introduced to make the system more precise and accurate. We also reviewed the existing post-processing techniques of sensor data which has a significant role to make a convenient joint monitoring system.

After evaluating all these technologies and methods, it seems that the development of a viable joint monitoring system can turn joint-related study (clinical and non-clinical) in a new direction. Wearable light-weight devices with miniature sensors and wireless connectivity appear to have the greatest potential in terms of ease of use, cost and the measurement of accurate and clinically relevant information. Researchers are now also focusing on noise reduction of the sensors, error minimization, and the handling of a large amount of data to build a secured and reliable device. Furthermore, the evolution of data science technologies has opened the door of possibility to develop and integrate an efficient prediction and feedback model with the system for more effectively taking care of the human joint health. Despite the enormous technological advances in the past few decades, their application in the real world is hindered by challenges that are present in those systems. That is why further research work and development are required to overcome those challenges by focusing on some key factors that include the following.
Most of the joint monitoring researches are focused on developing the sensing system using different technologies and sensing combinations. There is much less research emphasizing the data post-processing techniques and building predictive models. More work is needed to define an efficient prediction and feedback model depending on the properties of the data set and the experimental settings.The majority of the published studies employed different methods to assess validity and reliability which makes it difficult to compare the monitoring devices. In addition, clinical acceptance is questionable due to the lack of enough involvement of medical professionals during the design and evaluation process. Therefore, standard validation criteria and protocols (including clinical protocols) should be developed by the major regulatory bodies and clinical researchers to evaluate the accuracy and reliability of a monitoring device. These standards would provide guidelines to be used in the development and use of high-quality devices by both researchers and consumers.One of the main challenges is to extract and select features in real-time systems since the modeling techniques can handle the raw extracted features. This causes unnecessary redundancies which reduce the accuracy and efficiency of the system. This can be resolved by integrating cloud server communication with the system for real-time data mining. Therefore, the cloud server can handle all sets of data by using proper algorithms.The accuracy of joint health assessment using a monitoring device is heavily affected by the amount and variety of training data. It is preferable if the training data set contains data from as many subjects as possible. However, it is challenging to coordinate human subjects of different ages and musculoskeletal conditions to collect a large amount of joint monitoring data sets. This is a major barrier to evaluate the effectiveness of the monitoring devices, especially in expensive clinical trials settings.Although the sensor fusion technique is an advanced approach, very few research studies are conducted in the field of joint monitoring. The major challenges of using multiple sensors in one system are data acquisition and processing, simultaneous wireless communication and synchronization. To solve the communication problem, we need to install a wireless communication module which supports multiple connections for different sensors. We also need to calibrate all the sensors efficiently and use data standardization method to overcome the difficulties related to data processing and management. Moreover, the selection of proper sensor combination in a multi-sensor system is crucial to enhance the performance of a joint monitoring system.The hardware and computational resources for a monitoring system can be a crucial factor for long term communication and data acquisition. Therefore, high configuration hardware support is needed with an efficient algorithm which can deal with large data set resourcefully. On the contrary, more resource will consume more power which is one of the most critical factors to be considered while building a system. To develop a balanced system, the power requirement of the system should be minimized by selecting power-efficient components and more efficient power supply. Energy harvesting can also be an option to solve this problem.As the system requires processing and transmitting health information of users, information security is a key aspect to consider. It includes data privacy, security as well as ethical requirements recommenced by responsible regulation bodies. The scope of security and ethical requirements need to be clearly defined and specified. Besides, more efficient and secured algorithms are needed in order to ensure highly secured communication channels in existing low power, short range wireless platforms.Also, to obtain widespread acceptance among users, the systems need to be simple, wearable, easy-to-use, cost-effective, non-invasive, unobtrusive and inter-operable among various operating platforms. Therefore, more research and development efforts are needed to enhance the systems’ acceptance from both medical, user and business perspectives.Overall, a smart wearable joint monitoring and assistive device is expected to help the people at high levels of musculoskeletal health risk by tracking and assessing the joint function in a comfortable and non-intrusive manner. By integrating efficient prediction and feedback models, the system can be exploited for applications such as fall detection and prevention, athletes’ performance evaluation and rehabilitation progress. In rehabilitation from a joint injury, biofeedback is especially important since errors or mistakes in joint exercises can be corrected immediately, thus promoting faster and better recovery. Furthermore, the internet of things (IoT) is already unlocking the benefits of the advanced computing and communication technologies in the healthcare industry by connecting a variety of wearable healthcare systems. Hence, the development of a smart joint monitoring system coupled with IoT can facilitate remote long-term health monitoring and provide important joint-related information to medical professionals. This may result in early and accurate diagnosis of joint and joint-related problems and more efficient and effective medical intervention when needed.

## Figures and Tables

**Figure 1 sensors-19-02629-f001:**
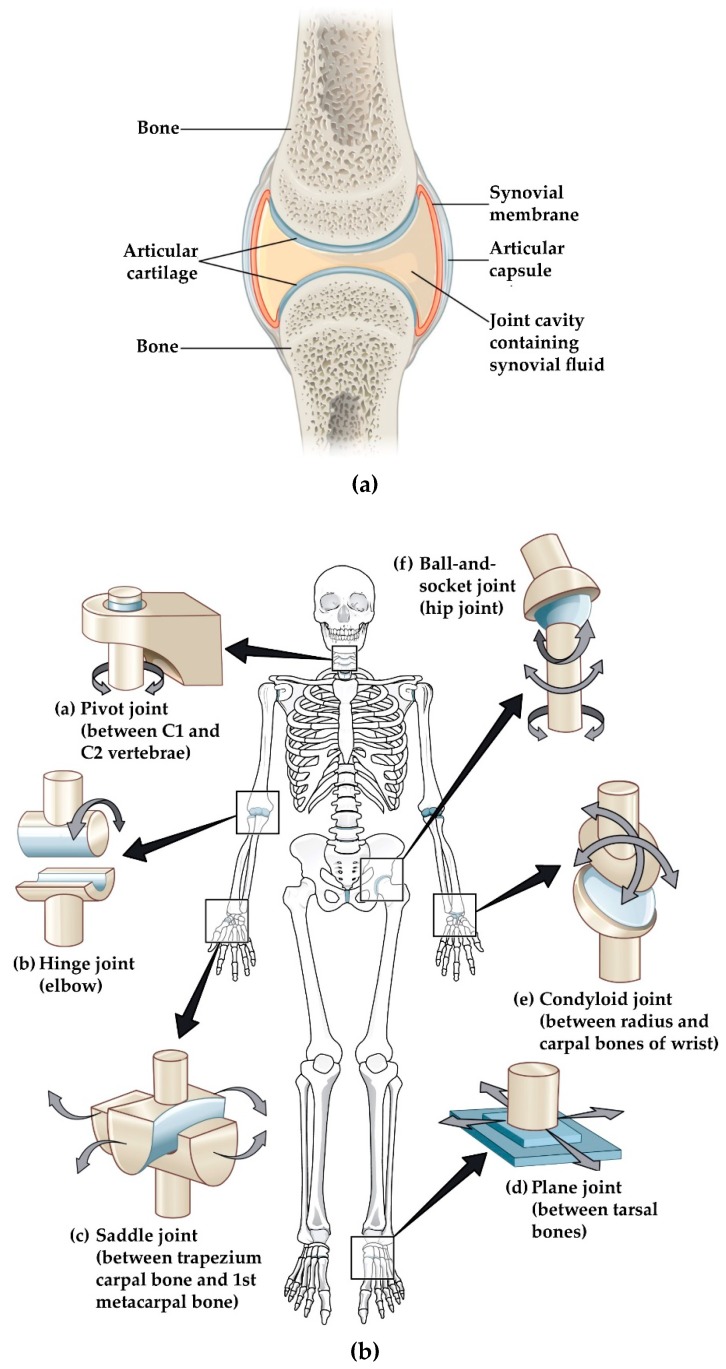
(**a**) Synovial joints; (**b**) Types of synovial joints. Image source: https://opentextbc.ca/anatomyandphysiology/chapter/9-4-synovial-joints/under a Creative Commons Attribution 4.0 International License.

**Figure 2 sensors-19-02629-f002:**
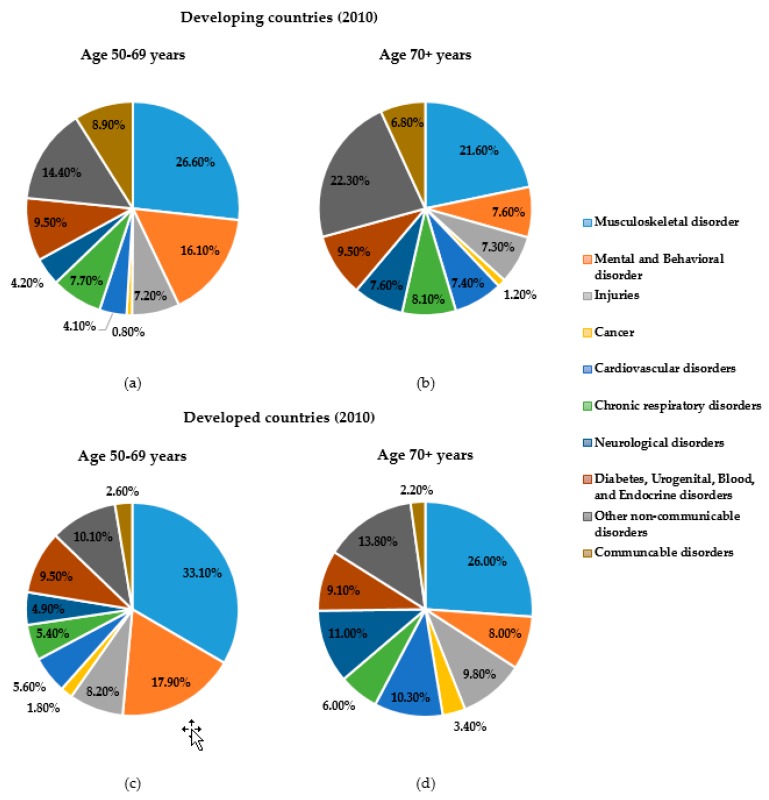
Proportion of total global years lived with disability (YLDs) in older age (50 years and more) groups attributable to each major set of health conditions in developing countries (**a**) 50–69 years; (**b**) 70+ years; and developed countries (**c**) 50–69 years; (**d**) 70+ years. Source: The Global Burden of Disease Study 2010 (GBD 2010).

**Figure 3 sensors-19-02629-f003:**
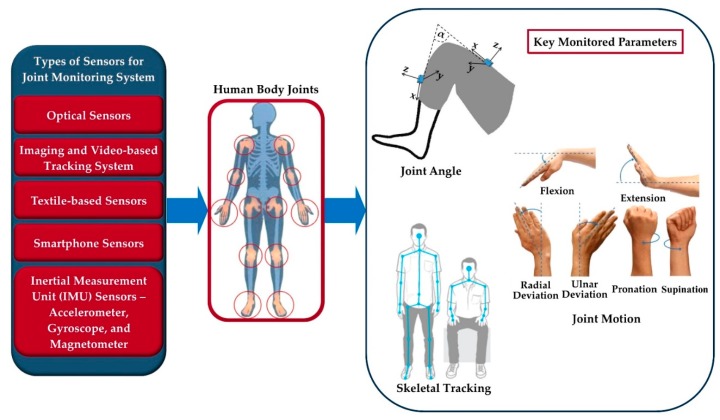
Joint monitoring sensor technologies and monitored parameters.

**Figure 4 sensors-19-02629-f004:**
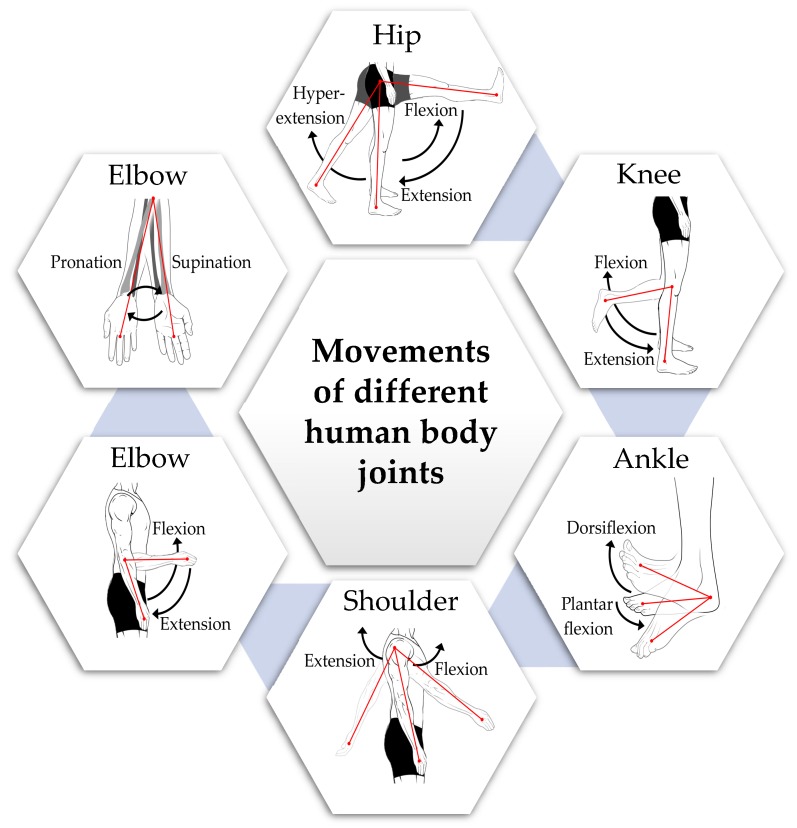
Different types of human body joints’ movements.

**Figure 5 sensors-19-02629-f005:**
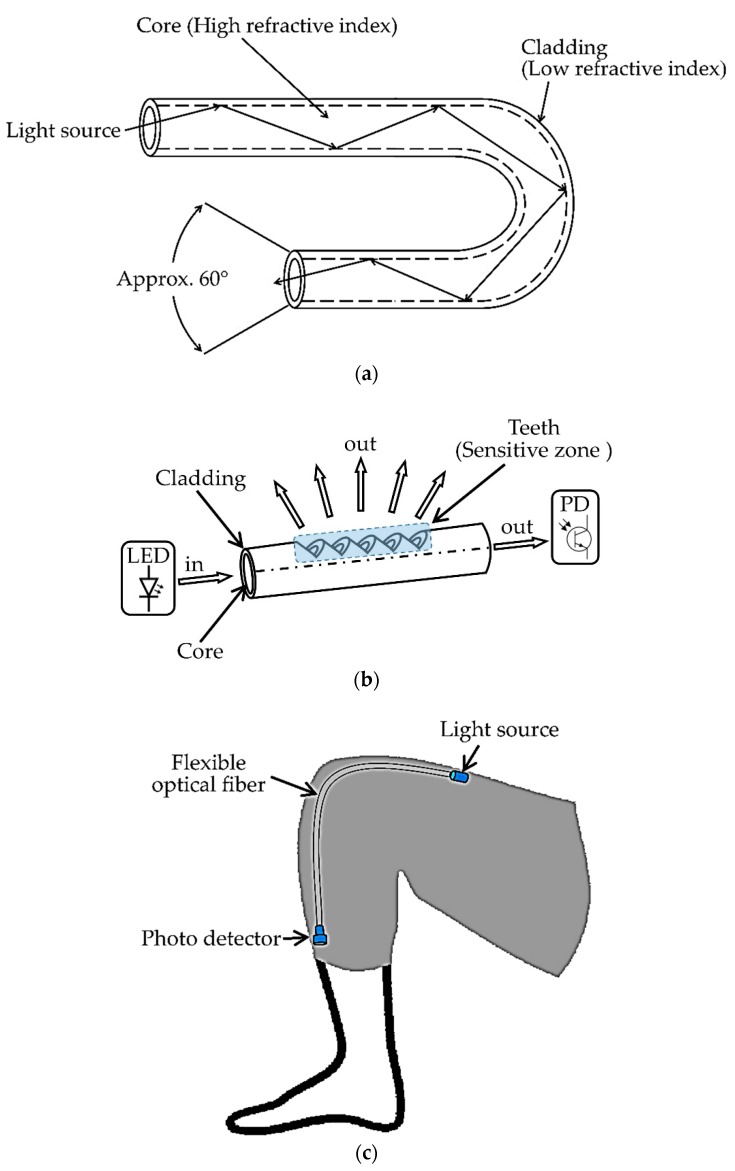
(**a**) Optical fiber configuration and working principle; (**b**) construction and operation method of the fiber-optic curvature sensor with a “teeth-like” sensitive zone; (**c**) flexible optical fiber sensors (OFS)-based joint monitoring system configuration.

**Figure 6 sensors-19-02629-f006:**
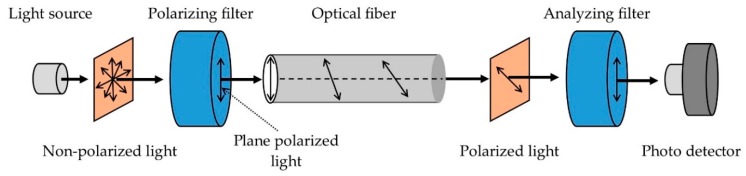
Components and working principle of optical-based goniometer system. The system was composed of five components: (1) a semiconductor laser as light source, (2) a Si p–i–n photodiode as photo detector, (3,4) two linear polarizers as polarizing and analyzing filters, and (5) a single-mode optical fiber as stress-induced birefringence polarization controller (SIBPC).

**Figure 7 sensors-19-02629-f007:**
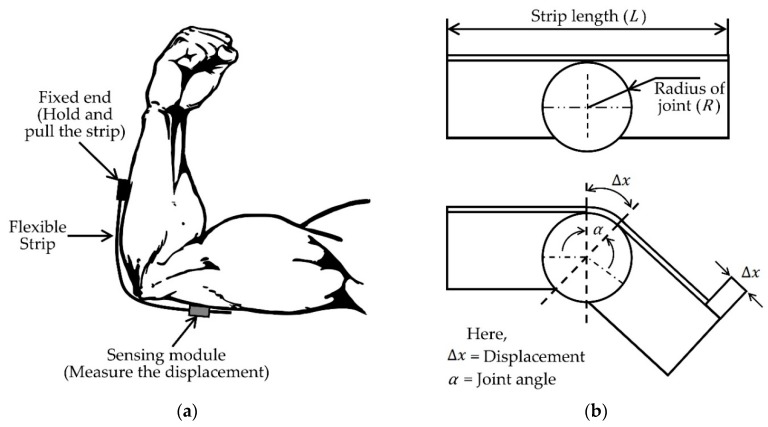
(**a**) Sensing setup of optical-based goniometer system for human elbow joint measurement; (**b**) operation method of the system.

**Figure 8 sensors-19-02629-f008:**
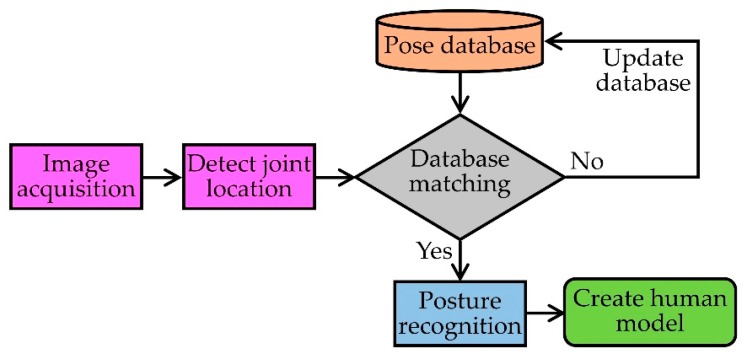
Block diagram of imaging-based human skeletal tracking.

**Figure 9 sensors-19-02629-f009:**
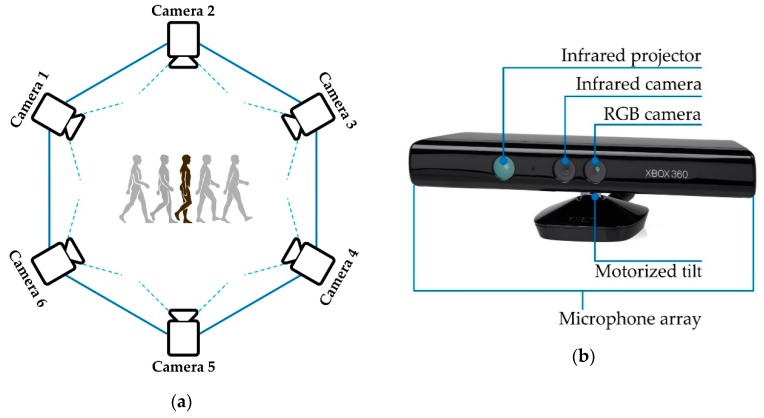
(**a**) Distributed camera networks for skeletal tracking; (**b**) Microsoft Kinect sensor system.

**Figure 10 sensors-19-02629-f010:**
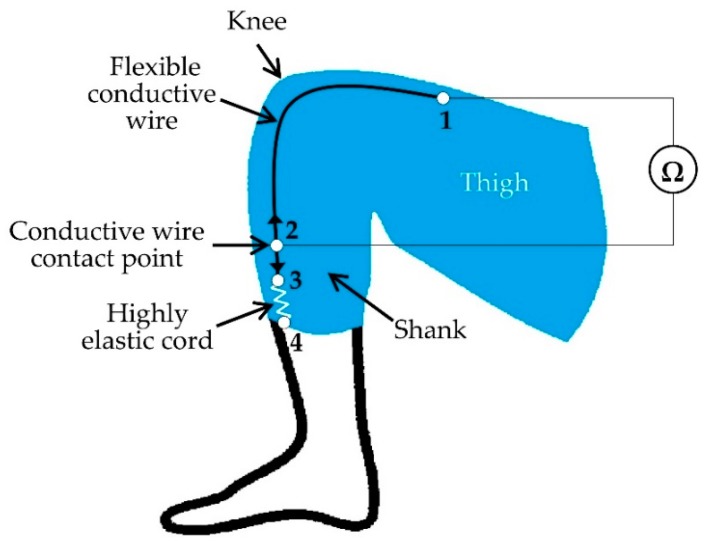
Schematic design of conductive wire sensor-based wearable joint monitoring device.

**Figure 11 sensors-19-02629-f011:**
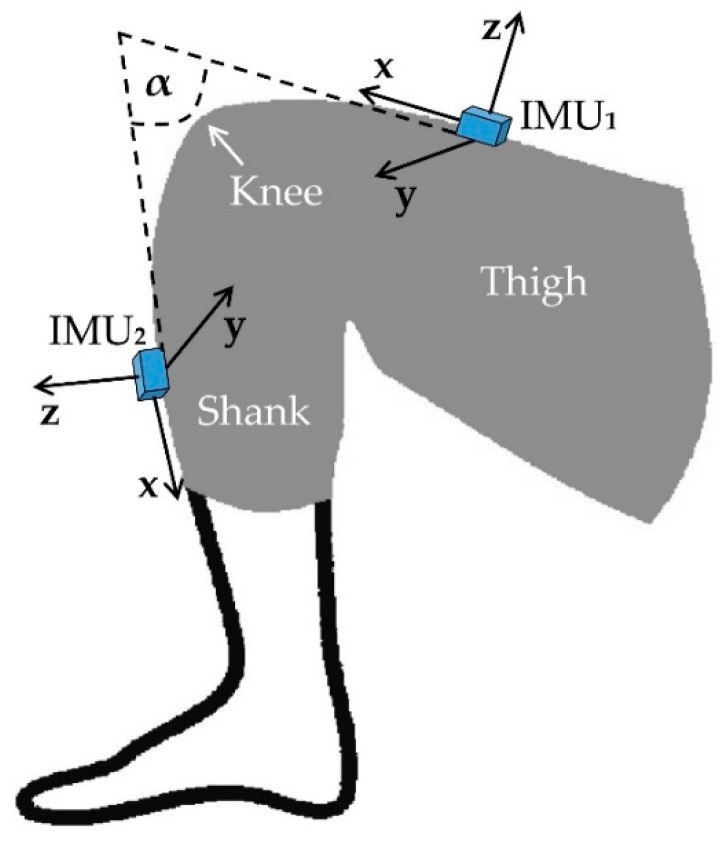
Inertial measurement unit (IMU) sensors’ orientation and position for knee angle measurement.

**Figure 12 sensors-19-02629-f012:**
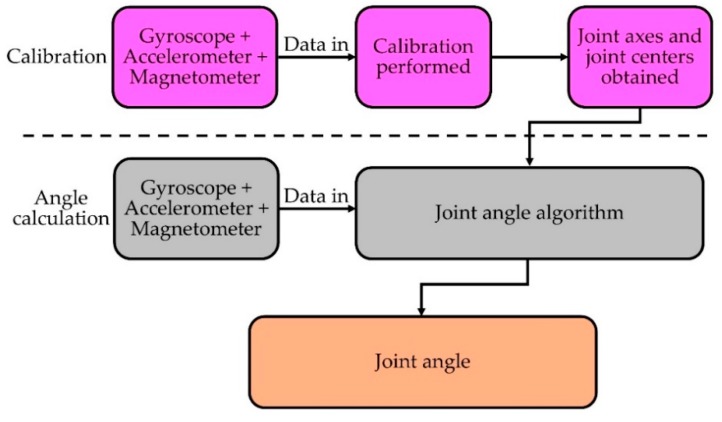
Process flow of joint angle measurement with IMU devices.

**Figure 13 sensors-19-02629-f013:**
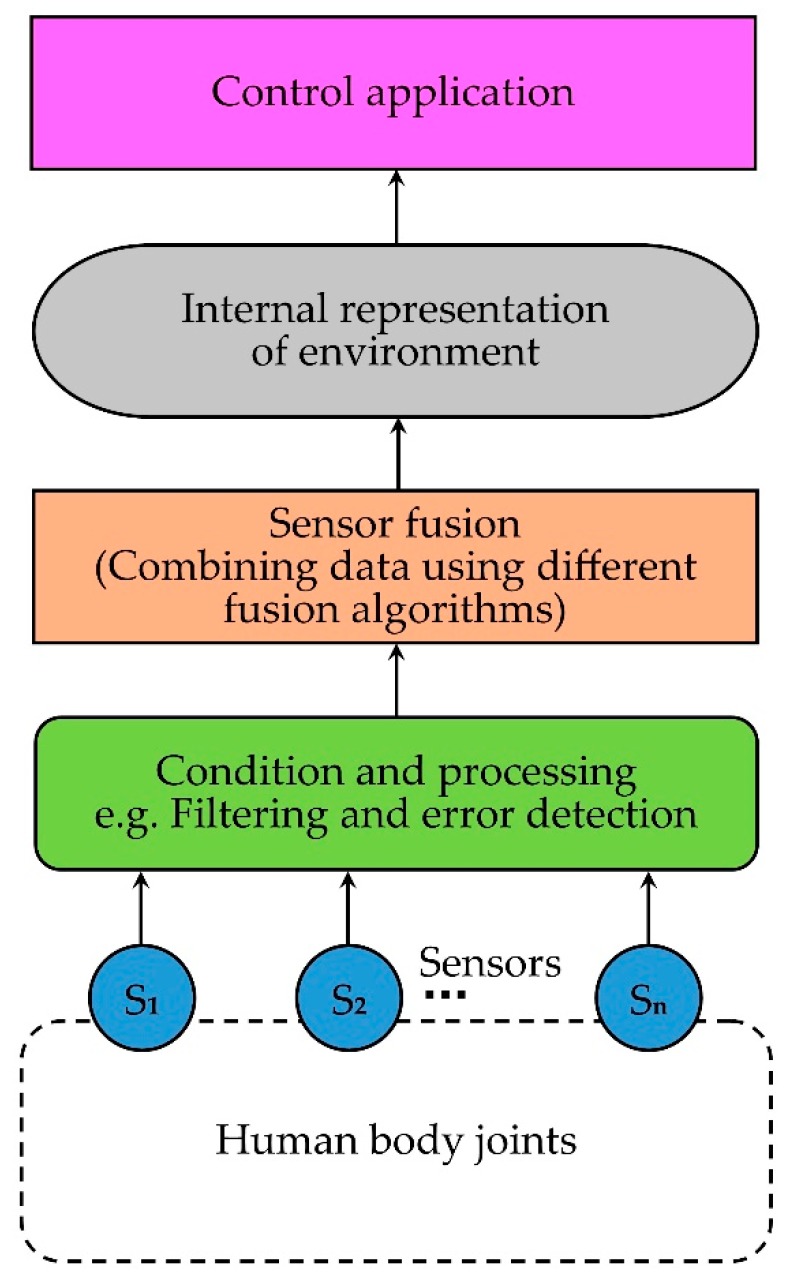
A simplified block diagram of sensor fusion methods.

**Figure 14 sensors-19-02629-f014:**
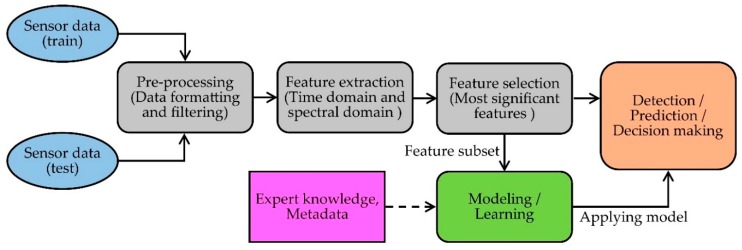
A generic flow diagram of a joint monitoring system.

**Table 1 sensors-19-02629-t001:** Types of synovial joints.

Joint Type	Joint Movement	Examples
**Pivot**	Rotation of one bone around another	Top of the neck
**Hinge**	Flexion/Extension	Elbow/Knee/Ankle
**Saddle**	Flexion/Extension/Adduction/Abduction/Circumduction	Thumb
**Plane**	Gliding movements	Inter-carpal/Tarsal bones
**Condyloid**	Flexion/Extension/Adduction/Abduction/Circumduction	Wrist
**Ball-and-socket**	Flexion/Extension/Adduction/Abduction/Rotation	Shoulder/Hip

**Table 2 sensors-19-02629-t002:** Comparison among different published sensor technologies for monitoring joints.

Ref.	Types of Sensor/Technology	Monitored Joint Parameters *	Measure	Method of Analysis	Advantages	Limitations
[25,26,27,28]	Optical fiber sensors	Angle	Attenuation of the transmitted optical signal power	Using the relation between the attenuation and the bending angle of the fiber	High resolution Flexibility Light-weight Long term reliability Immunity to electromagnetic interference	Limited measurement range (Angle) Nonlinearity Sensitive to temperature and humidity
[17,29,30]	Optical-based goniometer	Angle	Planar motion of an optical navigation sensor	Detecting navigation of the sensor using a miniature camera to calculate the bending of the joint	Compact and light-weight Flexibility High accuracy High speed of reaction	Sensitive to placement location May hinder natural joint movement during operation 3D sensing may not be possible
[31,32,33,34,35,36]	Imaging and video-based tracking system	Angle, motion, skeletal tracking	Visual data of several human actions	Skeletal tracking using anthropometric constraints and known joint locations in reference videos **	High accuracy and sensitivity Able to capture movements of multiple joints at a time No body-worn sensors are needed	Complex procedure with expensive infrastructure and sophisticated analyses Limited coverage area Requires body markers and adequate lighting condition for accurate measurements Unreliable to differentiate between near and far parts of human body, and for postures having self-occlusions ***
[37]	Textile-based conductive wire sensors	Angle	Changes of resistance	Changes of resistance are directly proportional to joint angles	Comfortable and suitable for long-term monitoring Simple mechanism One-time calibration Low-cost	Low resolution Low accuracy Nonlinearity Material uncertainties and hysteresis
[38,39,40,41,42]	Textile-based flex sensors	Angle	Changes of resistance	Changes of resistance are directly proportional to joint angles	Flexibility and stretchability Easily attachable with comfortable garments Low-cost	Fragile and lower lifetime (Prone to be damaged due to numerous bending) Low accuracy with noisy signal Nonlinearity Sensors are wide and affixing multiple sensors on the supportive garments is not feasible
[43,44,45,46,47]	Textile-based strain sensors	Angle, motion and rotation	Changes of resistance	Changes of resistance are directly proportional to joint angles and motion	Flexibility and stretchability High sensitivity Low-cost	Performance degradation due to large mechanical strains and rigorous deformations Signal drift due to the viscoelasticity of materials Limited to sense movements in the sagittal plane
[48]	Piezoresistive sensors – chopped carbon fiber (CCF)/polydimethylsiloxane (PDMS) yarns	Motion	Changes of resistance	Variation of relative resistance under mechanical deformation due to joint movements	Flexibility High sensitivity Easy integration into textile structures	Nonlinearity Material uncertainties and hysteresis Applying higher strain may cause piezoresistive performance (i.e., sensitivity) decay and delays the piezoresistivity transition
[49,50,51,52,53,54]	Smartphone sensors –accelerometer, gyroscope, magnetometer and camera	Angle, motion	Acceleration, inclination and camera measurements	Using smartphone applications to gather inbuilt sensors and camera data for measuring the range of motion	No external sensors are needed No external communication and data processing module are needed Applications are easy to implement	Lower accuracy comparing to other external sensors-based applications Difficult to place smartphones around different body joints Unable to monitor complex joint movements No standardized testing procedures are reported for clinical application
[55,56,57,58,59]	Acoustic emission (AE) sensors –piezoelectric-films/MEMS-based microphones	Angle, motion	High-frequency sound signal occurring during joint motion	Changes of surface resistance due to acoustic emission	Low-cost Light-weight Easy to attach around different body joints	High background and interface noise Nonlinearity Low accuracy
[60]	Gyroscope	Angle	Three axes angular rate	Joint angle is calculated by comparing the angular rate between two calibrated gyroscopes (below and above the joint)	Small size Low-cost Light-weight High resolution Easy to attach around different body joints	Produces some large drift over time Complex algorithms are needed to reduce noise and drift error At least two sensors are needed to measure accurate angle
[61]	Magnetometer	Angle, motion	Change of magnetic field	Change of magnetic field is directly proportional to joint motion	Feasible to measure complex joint angles Easy to control with digital circuits	Interference in the magnetic field by ferromagnetic and EMF-producing objects in the environment may decrease the accuracy of measurement Unreliable for detecting the orientations of joints in a 3D environment
[62,63,64,65,66,67,68,69,70,71,72,73,74,75,76,77,78,79,80,81,82]	Inertial measurement unit (IMU) sensors –accelerometer, gyroscope and magnetometer	Angle, motion, skeletal tracking	Three-dimensional acceleration, angular rate and the magnetic field vector	Three-dimensional angular velocities and linear accelerations are used to detect the position and orientation. Relative data from two calibrated IMUs are compared for tracking the joint angle and gait analysis	A combination of three sensors (Accelerometer, gyroscope and magnetometer) Compact and light-weight Small size Low-cost High resolution High accuracy Easy to attach around different body joints Built-in wireless module Built-in algorithms in new generation IMU sensors for calibration and to fix the sensors’ orientation with respect to a global fixed coordinate system Reliable for detecting the position and orientations of joints in a 3D environment	Sensors alignment is required in a multiple IMUs-based joint monitoring system Drift error from gyroscope (possible to compensate by fusing data from accelerometer and gyroscope)

* Joint angle: the angle between the two segments on either side of the joint; joint motion: the combination of the angle and the orientation of the joint; skeletal tracking: a technique to build a skeletal model of a human body by detecting the joint positions. ** Anthropometric constraints: size, shape and composition of the human body. *** Self-occlusion: one part of an object is occluded by another part from a certain viewpoint.

**Table 3 sensors-19-02629-t003:** Active range of motion (ROM) (°) for human joints by gender and age [92].

Age		2–8 years		9–19 years		20–44 years		45–69 years	
		Females (39)	Males (55)	Females (56)	Males (48)	Females (143)	Males (114)	Females (123)	Males (96)
**Joint Motion**	**Hip extension**	26.2 (23.9–28.5)	28.3 (27.2–29.4)	20.5 (18.6–22.4)	18.2 (16.6–19.8)	18.1 (17.0–19.2)	17.4 (16.3–18.5)	16.7 (15.5–17.9)	13.5 (12.5–14.5)
	**Hip flexion**	140.8 (139.2–142.4)	131.1 (129.4–132.8)	134.9 (133.0–136.8)	135.2 (133.0–137.4)	133.8 (132.5–135.1)	130.4 (129.0–131.8)	130.8 (129.2–132.4)	127.2 (125.7–128.7)
	**Knee flexion**	152.6 (151.2–154.0)	147.8 (146.6–149.0)	142.3 (140.8–143.8)	142.2 (140.4–144.0)	141.9 (140.9–142.9)	137.7 (136.5–138.9)	137.8 (136.5–139.1)	132.9 (131.6–134.2)
	**Knee extension**	5.4 (3.9–6.9)	1.6 (0.9–2.3)	2.4 (1.5–3.3)	1.8 (0.9–2.7)	1.6 (1.1–2.1)	1.0 (0.6–1.4)	1.2 (0.7–1.7)	0.5 (0.1–0.9)
	**Ankle dorsiflexion**	24.8 (22.5–27.1)	22.8 (21.3–24.3)	17.3 (15.6–19.0)	16.3 (14.9–17.7)	13.8 (12.9–14.7)	12.7 (11.6–13.8)	11.6 (10.6–12.6)	11.9 (10.9–12.9)
	**Ankle plantar flexion**	67.1 (64.8–69.4)	55.8 (54.4–57.2)	57.3 (54.8–59.8)	52.8 (50.8–54.8)	62.1 (60.6–63.6)	54.6 (53.2–56.0)	56.5 (55.0–58.0)	49.4 (47.7–51.1)
	**Shoulder flexion**	178.6 (176.9–180.3)	177.8 (176.7–178.9)	171.8 (169.8–173.8)	170.9 (169.1–172.7)	172.0 (170.9–173.1)	168.8 (167.3–170.3)	168.1 (166.7–169.5)	164.0 (162.3–165.7)
	**Elbow flexion**	152.9 (151.5–154.3)	151.4 (150.8–152.0)	149.7 (148.5–150.9)	148.3 (146.8–149.8)	150.0 (149.1–150.9)	144.6 (143.6–145.6)	148.3 (147.3–149.3)	143.5 (142.3–144.7)
	**Elbow extension**	6.8 (5.2–8.4)	2.2 (0.9–3.5)	6.4 (4.7–8.1)	5.3 (3.6–7.0)	4.7 (3.9–5.5)	0.8 (0.1–1.5)	3.6 (2.6–4.6)	-0.7 (–1.5–0.1)
	**Elbow pronation**	84.6 (82.8–86.4)	79.6 (78.8–80.4)	81.2 (79.6–82.8)	79.8 (77.8–81.8)	82.0 (81.0–83.0)	76.9 (75.6–78.2)	80.8 (79.7–81.9)	77.7 (76.5–78.9)
	**Elbow supination**	93.7 (91.4–96.0)	86.4 (85.3–87.5)	90.0 (88.0–92.0)	87.8 (85.7–89.9)	90.6 (89.2–92.0)	85.0 (83.8–86.2)	87.2 (86.0–88.4)	82.4 (80.9–83.9)

**Table 4 sensors-19-02629-t004:** Listing of published inertial measurement unit (IMU)-based joint monitoring techniques, analysis and validation methods.

Ref.	Year	Sensor Units and Module	Sampling Frequency	Wireless	Analysis (Joint)	Reference System and Validation
[64]	2008	2 (gyroscope + accelerometer)	240 Hz	No	Knee angle (3D)	Magnetic motion capture system RMS errors: 4° (flexion/extension) 5° (abduction/adduction) 10° (internal/external Rotation)
[68]	2009	2 (gyroscope + accelerometer)	240 Hz	No	Knee angle (3D)	Visual aligned IMU system Accuracy: between 4.0° and 8.1°
[63]	2011	2 (Gyroscope + Accelerometer + Magnetometer)	5 Hz	Yes	Knee angle	Infrared motion capture system Average deviation: 0.08° to 3.06°
[69]	2013	4 (gyroscope + accelerometer + magnetometer)	120 Hz	Yes	Knee angle for both prosthesis and the contralateral leg	Optical 3D motion capture system RMS errors: <0.6° (Prosthesis) <4.0° (Contralateral leg)
[70]	2013	2 (gyroscope + accelerometer + magnetometer)	128 Hz	Yes	Elbow, forearm and shoulder movement	Optical tracking system RMS errors: 6.5° (Elbow flexion/extension) 5.5° (Forearm supination/pronation) 5.5° (Shoulder flexion/extension) 4.4° (Shoulder abduction/adduction)
[74]	2013	2 (gyroscope + accelerometer + magnetometer)	Not mentioned	Yes	Knees, elbows, toes, hip, shoulder, wrist, ankle, neck, forearm and thumb joints	Not mentioned
[62]	2014	4 (gyroscope + accelerometer + magnetometer)	120 Hz	Yes	Knee angle for both prosthesis and the contralateral leg	Optical 3D motion capture system RMS errors: <1.0° (Prosthesis) <3.0° (Contralateral leg)
[71]	2015	3 (gyroscope + accelerometer + magnetometer)	10–100 Hz	Not mentioned	Hip and knee joint tracking	Optical tracking system RMS error: <3.0°
[67]	2016	4 (gyroscope + accelerometer + magnetometer)	50 Hz	Not mentioned	Gait analysis by monitoring hip, knee and ankle joints	A computer mathematical simulation, a universal goniometer system and a real gait test max RMS error: 1.70°
[82]	2016	4 (gyroscope + accelerometer)	40 Hz	Yes	Knee angle for estimating human movement	Goniometer-based system highest angle deviation: 2.0°
[72]	2016	1 (Gyroscope + Accelerometer)	100 Hz	Not mentioned	Hip and knee angles	A stereophotogrammetrical system RMS error: <3.2°
[66]	2017	2 (gyroscope + accelerometer + magnetometer)	30 Hz	No	Knee angle for human gait analysis	A vision-based motion capture system high correlation between two measurements (>0.947)
[65]	2017	2 (gyroscope + accelerometer)	128 Hz	Not mentioned	Knee angle, heel-strike and toe-off events for gait analysis	A commercial motion capture software RMS error: 8.0°
[73]	2017	2 (gyroscope + accelerometer)	30 Hz	Yes	Validation of a knee angle measurement	A DARwIn OP robot as ground truth system for knee angle measurement RMS error: <6.0° (When robot was walking) <5.0° (When robot kept the left leg stretched and performed an angle of −30°)

**Table 5 sensors-19-02629-t005:** A category of different feature selection methods, their advantages and limitations.

Methods		Advantages	Limitations	Example
**Wrapper methods**	Deterministic	- Simple - Dependence to feature - Interplay with classifier - Slower than Randomize	- High risk to over-fitting - More entrapment to local optimum than Randomize - Classifier dependent selection	- Sequential forward selection (SFS) - Sequential backward elimination (SBE) - Beam search
	Randomize	- Dependence to feature - Less entrapment to local optimum - Interplay with classifier	- Classifier dependent selection - More risk of over-fitting than deterministic	- Simulated Annealing - Randomized hill climbing - Genetic algorithms - Estimation of distribution algorithms
**Filter methods**	Univariate	- Quick - Gradable - No dependence to the classifier	- Relinquish dependence to feature - Relinquish interplay with the classifier	- Information Gain (IG) - x^2^ − CHI - t-test
	Multivariate	- Dependence to feature - No dependence to the classifier - Better time complexity than wrapper	- Slower than univariate methods - Less gradable than univariate methods - Relinquish interplay with the classifier	- Correlation-based feature selection (CFS) - Markov blanket filter (MBF) - Fast correlation-based feature selection (FCBF)
**Embedded methods**		- Dependence to feature - Interplay with classifier - Better time complexity than wrapper	- Classifier dependent selection	- Decision trees - Weighted naive Bayes - Feature selection using the weight vector of SVM

**Table 6 sensors-19-02629-t006:** Advantages and limitations of different classification models.

Methods	Advantages	Limitations
**k-Nearest Neighbor**	- Easy to understand and easy to implement - Training is very fast - Robust to noisy training data - It is particularly well suited for multimodal classes	- It is sensitive to the local structure of the data - Memory limitation - Being supervised learning lazy Algorithm e.g., runs slowly
**Neural Network**	- Efficiently handles noisy inputs - Computational rate is high - When an element of the neural network fails, it can continue without any problem with their parallel nature	- It is semantically poor - Difficult in choosing the type of network architecture - Requires high processing time for large neural networks
**Gaussian Mixture Model**	- Training is very fast - Performs well on the data of different size and densities	- The result is not stable - Sensitive to violations and distributional assumptions
**Hidden Markov Model**	- Convenient for modeling sequential data - Learning can take place directly from raw data	- Often has a large number of unstructured parameters - Unable to capture higher order correlation
**Decision Tree**	- Requires little data preparation - Nonlinear relationships between parameters do not affect tree performance - Easy to interpret and explain - Performs well with large data in a short time	- Complexity - Possibility of duplication with the same sub-tree on different paths
**Support Vector Machine**	- Produces very accurate classifiers - Less over-fitting, robust to noise - Especially popular in text classification problems where very high-dimensional spaces are the norm - Memory-intensive	- Requires both positive and negative examples - Needs to select a good kernel function - SVM is a binary classifier. To do a multi-class classification, pair-wise classifications can be used (one class against all others, for all classes) - There are some numerical stability problems in solving the constraint, QP (Quadratic programming) - Computationally expensive, thus runs slow
**Self-Organizing Map**	- Simple and easy-to-understand - A topological clustering unsupervised algorithm that works with a nonlinear data set - The excellent capability to visualize high- dimensional data onto 1 or 2-dimensional space makes it unique especially for dimensionality reduction	- Time consuming algorithm
**k-Means**	- Low complexity	- Necessity of specifying k - Sensitive to noise and outlier data points - Clusters are sensitive to the initial assignment of centroids
**Fuzzy Measure**	- Efficiently handles uncertainty - Properties are described by identifying various stochastic relationships - Allows a data point to be in multiple clusters	- Without prior knowledge, the output is not good - Precise solutions depend upon the direction of decision
**Expectation-Maximization Meta**	- Can easily change the model to adapt to a different distribution of data sets - Parameters number does not increase with the training data increasing	- Slow convergence in some cases
**Bayesian Classifier**	- Improves the classification performance by removing the irrelevant features - Good performance - Short computational time	- Information theoretically infeasible - Computationally infeasible

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
