# Peer review of "Monitoring Methods of Human Body Joints: State-of-the-Art and Research Challenges"

_sensors, 2019, doi:10.3390/s19112629_

Round 1

Reviewer 1 Report

The authors presented a review about Monitoring Methods of Human Body Joints. I confess that this is the first paper related to this field that I heard about.

The paper is well structured and presents and overview of the field. I recommend the authors to add an ": A survey" in the end of the title. 

Reviewer 2 Report

The document is well written and presents a survey on the subject, although not exhaustive, intended for non-experts. In fact, this paper is a survey of the different technologies and approaches proposed in the literature concerning the joint monitoring of the human body.
For this reason, the title and the abstract correspond poorly with the content of the paper. In fact, the authors have not developed any monitoring system, nor do they present a comparison between the performance of systems developed by others. The following title would have been more appropriate: "A survey on Monitoring Methods of Human Body Joints". In fact, the paper presents the state of the art of this type of monitoring. Finally, based on what the authors write in the introduction, I would have expected the advantages of wearable sensors to be presented, which are not described. On the contrary, the authors explain that the IMU sensors are not able to realize the skeleton traking. For the above reasons, the paper does not provide any truly significant contributions in the field. Even the references, although numerous, do not include many of the most important ones in the field.

Reviewer 3 Report

The aim of this study was to present a state-of-the-art survey on different monitoring methods of human body joints which includes the key joint parameters, sensor technologies and the developed systems and their performance analyses. The authors succeeded in writing a well-structured, clear, but still concise survey on the subject. I only have some minor remarks:

-      Continuous monitoring won’t improve mobility, as stated in the introduction. It will, however, make early diagnosis possible, which will eventually improve outcome. The authors have skipped a few steps in the treatment process here.

-      Always use capitals when referring to sections e.g. line 119.

-      Figure 2 does not add a lot to the numbers that were already given.

Round 2

Reviewer 2 Report

The authors have considered carefully all the comments of the reviewer and thoroughly revised the original submission, which now incorporates all responses to the reviewer’ annotations.